# Effects of Aluminium Contamination on the Nervous System of Freshwater Aquatic Vertebrates: A Review

**DOI:** 10.3390/ijms23010031

**Published:** 2021-12-21

**Authors:** Marie Closset, Katia Cailliau, Sylvain Slaby, Matthieu Marin

**Affiliations:** 1University Lille, CNRS, UMR 8576-UGSF-Unité de Glycobiologie Structurale et Fonctionnelle, F-59000 Lille, France; marie-closset@outlook.fr (M.C.); katia.caillau@univ-lille.fr (K.C.); 2Normandie University, UNILEHAVRE, CNRS, UMR 3730 SCALE, Environmental Stress and Aquatic Biomonitoring (SEBIO), F-76600 Le Havre, France; sylvain.slaby@univ-lehavre.fr

**Keywords:** aquatic contamination, aluminium, nervous system, development, *Xenopus*, zebrafish

## Abstract

Aluminium (Al) is the most common natural metallic element in the Earth’s crust. It is released into the environment through natural processes and human activities and accumulates in aquatic environments. This review compiles scientific data on the neurotoxicity of aluminium contamination on the nervous system of aquatic organisms. More precisely, it helps identify biomarkers of aluminium exposure for aquatic environment biomonitoring in freshwater aquatic vertebrates. Al is neurotoxic and accumulates in the nervous system of aquatic vertebrates, which is why it could be responsible for oxidative stress. In addition, it activates and inhibits antioxidant enzymes and leads to changes in acetylcholinesterase activity, neurotransmitter levels, and in the expression of several neural genes and nerve cell components. It also causes histological changes in nerve tissue, modifications of organism behaviour, and cognitive deficit. However, impacts of aluminium exposure on the early stages of aquatic vertebrate development are poorly described. Lastly, this review also poses the question of how accurate aquatic vertebrates (fishes and amphibians) could be used as model organisms to complement biological data relating to the developmental aspect. This “challenge” is very relevant since freshwater pollution with heavy metals has increased in the last few decades.

## 1. Introduction

Since the nineteenth century, anthropogenic activities have significantly altered ecosystems and triggered the sixth biodiversity crisis [1]. One of the main causes of this biodiversity erosion is the release of micropollutants of diverse nature and origins [2,3,4]. Once emitted, contaminants can reach non-target areas via several kinds of transport (e.g., runoffs, wet and dry deposition, long range transports), where they can have hazardous impacts on biodiversity. For example, the pollution of surface waters causes significant environmental and health issues [5,6,7].

Toxic metals, including aluminium (Al), negatively affect aquatic organisms [6,8,9]. Al is the third most common mineral and the most prevalent natural metallic element in the Earth’s crust, accounting for 8.1% of the Earth’s mass [10]. It naturally occurs exclusively in the +3-oxidation state (Al^3+^) in combination with other elements such as oxygen, silicon, and fluorine [11,12,13]. Al^3+^ is the major component of a large number of minerals, including mica, feldspars, and clays [12], and is naturally released into the environment through the weathering of rocks or minerals or through volcanic activities [13]. Produced by electrolysis from bauxite, Al is commercially manufactured under various forms, including particles in paints, pigments, and coatings, and it is used as a catalyst in the chemical and paper industries or textile dyeing. [13]. It has many industrial applications, particularly in electrical engineering, transportation, construction, and in the manufacture of household utensils, appliances, and packaging materials [11,13]. Aluminium sulphate (Al_2_(SO_4_)_3_) is widely used to improve the clarity of drinking water [14], and various Al compounds are used in processing, packaging, and the preservation of food [15]. In addition, Al has cosmetic and medical applications. It is found in antiperspirants, antacids and adjuvants for vaccines, toxoids, or used in patients with kidney failure to prevent hyperphosphatemia [16,17,18].

Due to the large number of natural and anthropogenic sources, Al is abundant in the environment. It has incompatible properties with fundamental life processes [12,19] and displays harmful effects in living organisms. In fact, Al is responsible for oxidative stress, cytotoxicity, genotoxicity, pro-inflammatory effects, immunological alterations, peptide denaturation or transformation, enzymatic dysfunctions, metabolic derangements, membrane disruption, microtubule perturbation, iron dyshomeostasis, amyloidogenesis, apoptosis, necrosis, and dysplasia [20]. Studies on animals have also shown that Al is neurotoxic and targets the central nervous system [11,19,21,22,23,24] by crossing the blood–brain barrier or by being transported through olfactory nerves [25]. In a rodent model, Al causes neurodegeneration, nerve cell death, changes in acetylcholinesterase (AChE) and neurotransmitter levels, histopathological changes (such as neuronal vacuolisation), and impaired cognitive and locomotor performances [12,24,25,26,27,28]. In humans, it is known to be associated with many pathologies of the nervous system, such as Alzheimer’s and Parkinson’s diseases, dementia, and autism [20].

While Al exposure is recognised to reduced survival, reproduction, and growth rates in fish and amphibians [10,29,30,31,32,33,34,35,36,37,38,39,40], only a few studies have addressed the neurotoxicity impacts in aquatic vertebrates. However, damages to the nervous system could alter the relational functions of organisms, threatening their survival, reproduction, and, ultimately, the population dynamics. Therefore, it appears of major importance to characterise the effects and action mechanisms of this contaminant on the nervous system of aquatic vertebrates. This literature review reports the effects of Al on the nervous system of freshwater aquatic vertebrates. It also poses the question of accurate aquatic vertebrates as model organisms that could complement the biological data relating to the developmental aspect.

## 2. Aluminium in Surface Freshwater

As a major constituent of the Earth’s crust, Al’s natural release into the environment exceeds those resulting from human activities [41]. However, its concentration in surface waters is increased by human activities, such as industrial and municipal discharges and Al_2_(SO_4_)_3_ is also found in drinking water [21,42,43].

Properties of Al in soil and water, such as persistence, mobility, chemical reactivity, and sorption dynamics, are governed by physicochemical and geological parameters, such as pH, temperature, organic matter, and suspended matters [44,45,46], which also directly affect its bioavailability. Dissolved Al concentrations in surface waters are highly variable and strongly influenced by the pH and the amount of dissolved organic matter (DOM) [11,44]. Al and its derivatives are poorly soluble in water at pH comprised between 6 and 8, which is the case for most natural surface waters [12]. Nevertheless, recent environmental monitoring campaigns revealed its occurrence at concentrations exceeding the World Health Organization and United States Environmental Protection Agency standards (0.2 mg·L^−1^). Indeed, in 2010, Al was found in rivers and lakes sampled all around the world at 1.2 ± 0.8 mg·L^−1^ (n = 9) and at 1.6 ± 1 mg·L^−1^ (n = 8), and it could reach a mean concentration of 3.1 ± 1.9 mg·L^−1^ in waterbodies (n = 5) sampled in Asia [47]. Usually, high concentrations in natural waters are only observed when the water pH is below 5. Therefore, concentrations in most surface freshwaters (e.g., ponds, lakes, and streams) with a pH greater than 5.5 are less than 0.1 mg·L^−1^ [13,21]. However, acidification of freshwater ecosystems leads to Al mobilization. Strong pH depressions have an anthropogenic origin, resulting from acidifying mine drainage, rain, and fertilisers [12,48], but can also be natural with snowmelt in spring or erosion caused by storms [28,49]. In sulphide-rich regions, water is strongly acidic (pH less than 3.5) and soluble Al concentrations are close to 50 mg·L^−1^ [21] and can reach 90 mg·L^−1^ due to acid mine drainage and discharge [46]. Additionally, in urban and industrial areas, high concentrations are regularly quantified [13].

Variations in Al toxicity are also observed according to pH or DOM fluctuation. When the pH is below 5.5, exposure to low concentration of Al (0.0125 mg·L^−1^) causes severe physiological alterations in *Oncorhynchus mykiss* [29]. In *Danio rerio*, exposure to 0.05 mg·L^−1^ of aluminium sulphate increases AChE activity in the brain at pH 5.8 but not at pH 6.8 [50]. Similarly, waters with high contents of DOM, such as humic or fulvic acid, promote the dissolution of Al and its derivatives (aluminium oxide and aluminium salts) [21]. Basically, DOM increases Al solubility while decreasing its toxicity [44]. High levels of suspended particles, which can be caused by storms, also strongly modify Al concentrations in surface waters by making new sites of adsorption available [51].

The behaviour of Al in aquatic ecosystems is strongly influenced by its binding chemistry. It can be suspended or dissolved as a monomer or a polymer, in the form of a free ion, and complexed to water molecules or bound to organic or inorganic ligands and negatively charged functional groups on humic materials and clay [46]. Different salts of Al can be found: aluminium oxide, chlorohydrate, hydroxide, fluoride, chloride, sulfate, lactate, phosphate, and nitrate [44,46]. Aluminium hydroxide and aluminium fluoride are the most important inorganic species in natural waters, although aluminium phosphate may be important for aluminium-treated wastewater [52]. Except for aluminium phosphide, anionic components (e.g., fluoride, chloride, and nitrate) do not affect the toxicity, although they affect the bioavailability [21]. The toxicity is decreased in ligands—complexed forms such as organic acids, fluoride, sulphate and silicate—and solely the monomeric inorganic form contributes to acute toxicity [45].

Aluminium speciation depends on several factors, including concentrations of dissolved organic carbon, fluoride, sulphate and phosphate, suspended particles, and water temperature and pH [44,53]. All parameters significantly alter its bioavailability and toxicity [53]. As previously specified, the toxicity increases at low pH (5.5) due to changes in speciation [45]. In water, for acidic pH values below 4, the dominant speciation corresponds to the oxidation state Al^3+^ and is generally in the form of a hydrated complex, Al(H_2_O)_6_^3+^. For a pH between 5 and 6, the Al_2_(OH)_2_^4+^ and Al(OH)_5_^2−^ species dominate, and Al may complex with phosphate and no longer be available. The insoluble form Al(OH)_3_ is a predominant form in the pH range between 5.2 and 8.8. Above pH 9, the soluble species Al(OH)_4_^−^ is dominant and is the only one present at pH levels above 10 [21,45]. At basic pH and under non-equilibrium conditions, Al polymerises and forms Al_2_(OH)_2_(H_2_O)_8_^4+^ and Al_13_(OH)_32_^7+^ polycations [46]. These structures become large enough to precipitate and carry Al, reducing its mobility. In general, monomeric Al compounds are more reactive and labile than polymeric compounds. However, the above considerations are only valid when the organic matter and silica contents remain low [13]. In the presence of large amounts of DOM, particularly fulvic acid, Al binds to these substances and becomes a dissolved organic complex [21,46].

Al bioconcentration in aquatic organisms, studied in fish and amphibians [33,39,54,55,56,57,58], also depends on several parameters, including pH and organic carbon content. For example, *Salvelinus fontinalis* accumulates more Al at pH 5.3 than at pH 7.2 [33]. In freshwater ecosystems, toxic metals, including metalloids, are widely sorbed on surface sediments and suspended particles that modulate their speciation, dispersion, and ecotoxicology [53,59]. Since many freshwater organisms are in contact with dissolved and particulate matter fractions, they accumulate Al from both water and solid phases [53], despite the bioaccumulation potential appearing low [12].

## 3. Effects of Aluminium on the Nervous System of Freshwater Aquatic Vertebrates

The neurotoxic action of Al impacts motor and cognitive capabilities. At the cellular level, several important mechanisms are affected: axonal transport, neurotransmitter synthesis, synaptic transmission, calcium homeostasis, energy metabolism, inflammatory responses, cell death, and glial cell activation [27]. At the molecular level, serious modifications occur in protein phosphorylation/dephosphorylation and degradation, gene expression, DNA repair, formation of reactive oxygen species, antioxidant enzyme activity, NF-kB and JNK pathways, and DNA binding [27]. However, these changes are essentially observed in mammals, and only a few studies have addressed the effects produced on the nervous system of aquatic vertebrates. Table 1 and Figure 1 report the effects of Al on the nervous system of several aquatic vertebrates.

The nervous system of aquatic vertebrates can accumulate Al, as proven in various species. For example, accumulation was observed in the brain of *Cirrhinus mrigala*, *Ctenopharyngodon idella*, and *Oncorhynchus mykiss* exposed to Al_2_(SO_4_)_3_ [55,58,66]. For *C. mrigala*, this observation could be due to a dysfunction of the liver, and thus, of a detoxification process, where Al was also detected [58]. In *O. mykiss* exposed to environmental concentrations, small deposits on the apical surface of the cerebrovascular endothelium and in the telencephalon indicated that Al crossed the blood–brain barrier. In the telencephalon, it was intimately associated with the membrane of neuronal cell bodies in the form of diffuse deposits surrounding the brain capillaries. In addition, cell bodies contained several distinct types of neural debris [55]. Accumulation is regulated by absorption and excretion rates, toxicant concentration, and exposure duration [54,58]. For instance, in *C. mrigala*, the rate of absorption and the biomagnification factor was higher, while the rate of excretion was lower in chronic compared to acute exposures [58]. In contrary to the previous studies, Anandhan and Hemalatha [54] did not detect Al accumulation in the brain of *D. rerio* exposed to 5.69 and 17.08 ppm of AlCl_3_, while accumulation occurred in the liver, gills, and muscles.

As shown in Table 1, most studies focused on the assessment of oxidative stress (which results in high production of free radicals) of Al on the nervous system. Al replaces iron in various biomolecules and increases intracellular iron concentrations, promoting a Fenton oxidation reaction [71,72]. Additionally, it disrupts the electron transport chain in mitochondria [66] and generates oxidising radicals. Oxidative stress is deleterious to organisms because it leads to protein and enzyme inactivation, lipid peroxidation, and DNA damages. Fish nervous tissue is particularly sensitive to Al-induced oxidative actions because of its richness in polyunsaturated fatty acids and high consumption of oxygen (about 1/5 of the total consumption) [62]. Therefore, oxidative damages of the nervous tissue are one of the main mechanisms leading to the toxic effects of Al [62]. In *Lepomis gibbosus*, oxidative stress occurred in nerve tissue [70]. In *D. rerio*, *Lepomis macrochirus*, *Rutilus rutilus*, *Carassius carassius*, and *Neogobius fluviatilis*, a significant increase in the level of brain lipid peroxidation was seen after exposure to AlCl_3_ [70,73]. This accumulation was also observed in *C. idella* exposed to Al_2_(SO_4_)_3_ [66] and in *Oreochromis mossambicus* after exposure to Al oxide nanoparticles (Al_2_O_3_NPs) [67].

Al-induced oxidative stress alters the activity of antioxidant enzymes. The enzyme activity, initially increased to compensate for the oxidative stress, is depleted by extended exposure, leading to protein and DNA damages [66]. A significant decrease in the brain catalase (CAT) activity was observed in *Channa punctatus*, *C. idella*, and *O. mossambicus* exposed to AlCl_3_, Al_2_(SO_4_)_3_, and Al_2_O_3_NPs, respectively [61,66,67]. The decrease was linked to the production of glutathione peroxidase (GPx), an antioxidant enzyme, in competition with CAT for the common hydrogen peroxide (H_2_O_2_) substrate [74] and with the establishment of non-enzymatic mechanisms as, for example, the sequestration of oxidant radicals by metallothioneins [75]. Another explanation is the inhibition of CAT by an Al ion capable of binding the enzyme thiol groups [66]. Additionally, the decrease in CAT activity could be explained by a decrease in gene expression [76]. Finally, antioxidant enzymes may themselves undergo oxidative changes [67,77]. In *D. rerio*, CAT activity increased significantly in the brain after long-term exposure to AlCl_3_. This increase reflects the need for a greater amount of antioxidant enzymes to eliminate free radicals produced during Al long-term exposure [61], as CAT. The activity of the superoxide dismutase (SOD) which neutralises oxidising radicals and converts superoxide ions [78] into H_2_O_2_ [66], is altered by aluminium exposure. On one hand, a significant increase in SOD activity was detected in the brain of *L. macrochirus*, *R. rutilus*, *C. carassius*, and *N. fluviatilis* after exposure to AlCl_3_ [62] and *C. idella* exposed to Al_2_(SO_4_)_3_ [66]. On the other hand, in *O. mossambicus* exposed to Al_2_O_3_NPs, a significant decrease in the brain of SOD, GPx, and glutathione S-transferase (GST) activities and an increase in the level of H_2_O_2_ were observed by Vidya and Chitra [67]. The decrease in SOD activity may result from the generation of an excess of oxidising radicals following the exposure to nanoparticles, which could then lead to inactivation of the enzyme. The oxidant radicals would further decrease the activity of other antioxidant enzymes, such as CAT, GPx, or GSR, decreasing the neutralization potential of oxidant radicals and increasing lipid peroxidation [67,77]. Finally, GST, another antioxidant enzyme also involved in tissue protection from oxidative stress and damages [79], increases its activity in response to a rise in free radicals [67]. It results in a GSH decrease, which normally acts as a GST cofactor to neutralise oxidising radicals [80]. A significant rise in GST activity and a decrease in reduced glutathione (GSH) content were generated in *O. mossambicus* exposed to Al_2_O_3_NPs [67].

Another harmful effect of Al on the nervous system of aquatic vertebrates is the alteration of AChE activity, a key nervous system hydrolase that catalyses the hydrolytic metabolism of the neurotransmitter acetylcholine (ACh) into choline and acetate [67]. AChE is usually used as a biomarker of effects for the central nervous system [81]. In fish, AChE activity is essential for muscle function and behaviour [82]. A significant increase in the enzyme activity was observed in *D. rerio* following exposure to AlCl_3_ [50,61] and in *Oreochromis niloticus* after exposure to AlCl_3_ and Al_2_(SO_4_)_3_ [65]. According to Maheswari et al. [61], AChE increased in activity in *D. rerio* after short exposure times and in quantity for longer exposure times. The increased activity could be due to an allosteric interaction between the anionic peripheral site of AChE and Al^3+^ ions [83], an increase in the production of free radicals [61,77], or a conformational change consecutively to the peroxidation of the membrane lipids of the brain cells [84]. In contrast, a significant decrease in AChE activity was observed in *O. mossambicus* following exposure to Al_2_O_3_NPs [67]. Al neurotoxicity also results in altered levels of brain neurotransmitters. In *C. idella* exposed to Al_2_O_3_NPs, a significant increase in dopamine and noradrenaline content was observed by Fernández-Dávila et al. [66], while the adrenaline content significantly decreased. The observed changes in these neurotransmitter levels could be related to their synthesis. These three neurotransmitters are derived from tyrosine. Dopamine is converted into noradrenaline, which is further converted into adrenaline. Enzymes catalysing these transformations are probably affected by the binding of Al to the thiol groups [66]. As mentioned previously, the induction of oxidising radicals may never be responsible for direct damages on the enzymes or indirect actions on the corresponding genes (in *C. idella* brain, [85]). The synthesis of catecholamines, which include dopamine, noradrenaline, and adrenaline, is sequential, and inhibition of the final stages probably increases the content of noradrenaline and dopamine, as seen in *C. idella* [66].

At the genetic level, a decrease in the production of NeuroD_1_ mRNA, involved in the regulation and the control of nerve differentiation, was observed in *Salmo salar* exposed to AlCl_3_ and was probably due to an increased level of stress [63,86]. Additionally, chromatin and DNA are particularly vulnerable to Al^3+^ [87]. Al ions strongly bind DNA, RNA, and mononucleotides [12,88,89]. In *L. macrochirus, R. rutilus, C. carassius*, and *N. fluviatilis*, exposure to AlCl_3_ induced an overexpression of glial fibrillary acidic protein (GFAP), a subunit of the cytoskeleton intermediate filaments, and S100β, a calcium-binding protein mainly present in astrocytes [62,90]. This overexpression correlated with an increase in the content of the lysed forms of GFAP and S100β fragments. This indicates that Al ions could activate intracellular proteases which alter intermediate filaments in astrocytes [62], as in *D. rerio* exposed to AlCl_3_ [37]. The overexpression of GFAP and S100β may be responsible for astrogliosis in *N. fluviatilis* exposed to AlCl_3_ [62] and in *L. gibbosus* [70]. Astrogliosis are changes characterised by an overexpression of GFAP, that occur in astrocytes in response to central nervous tissue injuries and damages induced by toxic substances in the brain of many vertebrates [62]. This glial cell reactivity is commonly used as a biomarker to detect nerve tissue disorders [90].

Morphological changes in tissues are considered as signs of various pathologies. In aquatic ecosystems, chronic exposure to contaminants at sublethal concentrations can alter the structural architecture of tissues without killing fish. Such structural tissue changes were observed by Vidya and Chitra [68] in the brain of *O. mossambicus* exposed to 4 mg^−1^ of Al_2_O_3_NPs (sublethal concentration). After 96 h of exposure, moderate degenerative changes occurred in all cerebral regions associated with a slight vacuolisation in the neural cells. After 60 days of treatment, severe degenerative changes and intracellular oedema were noted. As previously mentioned, Al_2_O_3_NPs can cross the blood–brain barrier, accumulate in nerve tissue, and induce damages to the brain [68]. These results are therefore in agreement with results obtained in *C. idella* by Sivakumar et al. [58] and in *O. mykiss* by Exley [55]. Vidya and Chitra [67,68], showing that deleterious effects of Al_2_O_3_NPs in *O. mossambicus* are persistent after cessation of exposure, indicating the irreversible neurotoxic properties of Al nanoparticles.

Finally, several studies have highlighted the impacts of Al on the behaviour of aquatic vertebrates in connection with an alteration of the nervous system. In *D. rerio* exposed to AlCl_3_, a significant decrease in the locomotor activity was demonstrated. A decrease in the distance travelled, a reduction of the maximum speed, and an increase in the absolute angle of rotation were mentioned [50]. The involvement of the cholinergic system in the locomotor activity, the response to new stimuli, and the performance of spatial memory tasks was fully established [91]. This implies that the induction of AChE activity in the brain observed in *D. rerio* may be responsible for the behavioural and neurotoxic effects of Al on the central nervous system [50]. Fish activity may also be limited by their compromised ability to extract oxygen from water. Al is believed to interfere with oxygen supply to tissues by causing osmoregulatory and ion-regulatory dysfunction and changing the haematological status [92,93]. In *D. rerio* larvae exposed to AlCl_3_, Capriello et al. [60] observed a significant decrease in the average of moved distance, velocity, time of movement, and number of heading at low concentrations (below 100 µM), with a recovery at high concentrations (100 and 200 µM). The impairment of the swimming ability of *D. rerio* larvae was probably caused by a reduction of the number of neural stem cell—limiting neuroblast differentiation [94]—and/or alteration of the glucose metabolism [95]. In *S. salar* exposed to AlCl_3_, Grassie et al. [63] observed an increased number of errors made by individuals in a maze, indicating a decrease in their spatial learning capabilities. Cognitive deficits are associated with a decrease in neuronal plasticity of the forebrain, and in NeuroD_1_, by an mRNA expression in the telencephalon [63]. Laming et al. [69] showed that a topical application of Al(OH)_3_ on the telencephalon of *R. rutilus* induces unusual and gentle lateral undulations of the body and a sporadic, violent, uncoordinated motor activity. These effects were associated with a delayed habituation of arousal responses to repeated presentations of two stimuli and the presence of electroencephalographic seizures in which the EEG amplitude was elevated from 4–20 times compared to a normal level. Even though *R. rutilus* lacks a cerebral cortex and has a relatively undifferentiated telencephalon, observed seizures are an expression of the malfunction of a fundamental mechanism, as in other vertebrate brains. Seizures correlate with over-activity of the brain, which normally operates during arousal [69]. The same topical application of Al(OH)_3_ in *Carassius carassius* induced a delayed habituation of cardiac arousal to a moving shadow stimulus [69]. Finally, Andrén et al. [96] showed that the swimming behaviour of the moor frog *Rana arvalis* is disturbed by environmentally relevant concentrations of AlCl_3_, while the behaviour of the common frog *Rana temporaria* and the agile frog *Rana dalmatina* exposed to the same concentrations are not. Altogether, aquatic vertebrates’ behaviour changes have serious consequences: limited survival in the wild [64] and affected swimming activity, predation, migration, and reproductive success [92].

## 4. Perspectives: Interests of Biological Models to Study the Effects of Aluminium on the Nervous System of Aquatic Vertebrates

Due to the permeable properties of the blood–brain barrier, the central nervous system is one of the major targets of Al in freshwater species. Several questions remain concerning the doses, the exposure times, and the sensitivity of the embryo developmental stages required to trigger a toxic effect. To date, no data exist on Al accumulation and cumulative/additive effects during specific periods of the neural system development. Using aquatic vertebrate models to perform dose–response tests, time-lapse exposures, and behavioural assessments in early developmental stages could provide precious pieces of information on Al toxicity. Studies on critical exposure phases could be precisely determined for the development of the central nervous system. For instance, the different phases of the neural embryonic development ranging from the early neural plaque induction and tube folding to the late formation of the neurogenic territories of the brain regions could have different sensitivity and accumulation rates. Moreover, these developmental parameters could generate important data and lead to the determination of sensitive toxicity periods and specific markers. There are advantages to using organisms such as *D. rerio* or *Xenopus sp.* in environmental toxicology studies [97,98,99]. Both models share a short life cycle [100,101], which can be studied from the oogenesis period to the late development in controlled conditions. The developmental stages, molecular signalling, genetic compositions, and the neurodevelopmental processes of both models are well characterised [102,103]. Both embryo nervous systems are visible by transparency and easily accessible for various in vivo recordings and studies [104]. Live-imaging at high resolution with structural and dynamical details and quantification of neuronal properties are also possible [105]. The neurotoxicity endpoints can be assessed during the neural development with proteomic and genomic large-scale screenings [106]. Their cellular and molecular neuronal parameters can be analysed in relation to the behavioural abnormalities, including locomotion, foraging, and avoidance [105,107]. Given the environmental concerns related to Al, its underestimated neurotoxic impacts on freshwater organisms, and also the interesting possibilities offered by the methods widely used on well-known fish and amphibian models, additional studies would allow a better understanding of the action of Al on the neural system and, more globally, its effect at the population level.

## 5. Conclusions

Al is responsible for various toxic effects. This metal is well-known for its neurotoxicity in mammalian models, but only a few studies have been conducted on aquatic organisms. However, due to the large number of natural and anthropogenic sources, Al is abundant in the environment and can be found in aquatic ecosystems. Previous works have shown that Al accumulates in the nervous system of freshwater vertebrates, where it can trigger oxidative stress, alter enzymatic activities, and neurotransmitters levels but also affect gene expression, cause astrogliosis and morphological changes, and impair behaviour and cognitive abilities. These effects were primarily studied in adult organisms without considering early stages of development, which are critical windows of exposure. In conclusion, further studies are needed to better characterise Al neurotoxic effects during whole developmental processes with the determination of the critical periods of time, duration, and the quantities that threaten freshwater life. Thus, Xenopus and Zebrafish could be valuable model organisms since their development are external and easily accessible. Sequential and additive exposures could be undertaken to understand the toxic mechanisms of the action of aluminium on the embryonic development of the nervous system and propose molecular signatures associated with functional states of media contaminated by this metal.

## Figures and Tables

**Figure 1 ijms-23-00031-f001:**
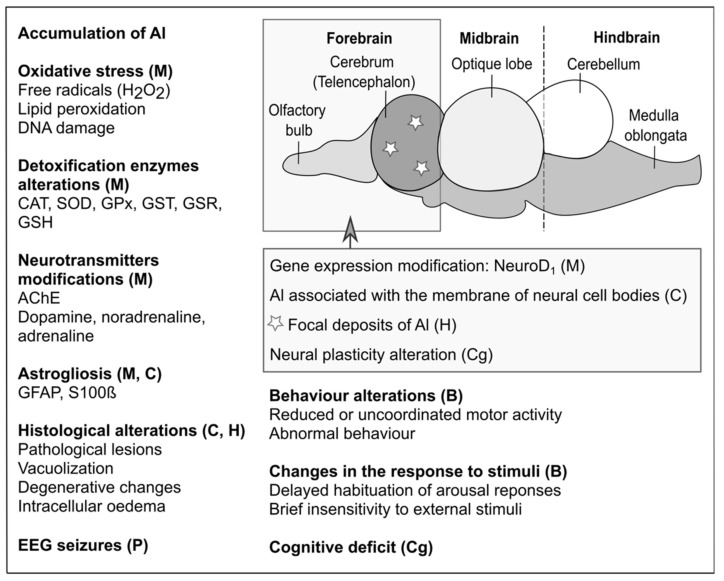
Molecular targets and alterations produced by aluminium in freshwater aquatic vertebrates. M: molecular effects. C: cellular effects. H: histological effects. P: physiological effects. B: behavioural effects. Cg: cognitive effects.

**Table 1 ijms-23-00031-t001:** Effects of aluminium on the nervous system of freshwater aquatic vertebrates reported in the literature.

Al Form	Species	Exposure Conditions	Effects	Ref.
AlCl_3_	*Danio rerio*	*In vivo*, embryos (6 hpf) to larvae (78 hpf) *Conc.:* 50, 100, 200 mM (sublethal conc.) *Duration:* 72 h	<100 mM: Significant ↘ in the average moved distance, velocity, time of movement, and number of heading 100 and 200 mM: Recovery below the control condition	[60]
AlCl_3_	*Danio rerio*	*In vitro*, embryos (4 hpf) to larvae (48 hpf) *Conc.:* 100 µM 44 h	↘ in the number of cells containing GFAP (marker of astroglia, a cell type involved in detoxification and stress defence) in the brain encephalon at the ventricular and subventricular levels and in the number of forebrain positive cells (GFAP-positive cells/total cells × 100) from 59.6% to 34.5%	[37]
AlCl_3_	*Danio rerio*	*In vivo*, adults *Conc.:* 150 ppm *Duration:* 7, 14, 21 d	In brain, ≥7 d: Significant ↗ in AChE activity 14 d: Significant ↗ in protein content 21 d: Significant ↘ in protein content ≥7 d: Significant ↗ in lipid peroxidation 21 d: Significant ↗ in CAT activity 7, 21 d: Significant ↗ in GST activity and in GSH content	[61]
AlCl_3_	*Danio rerio*	*In vivo*, adults (6–8 months) *Conc.:* 50 µg·L^−1^ *pH:* 5.8, 6.8 *Duration:* 24 h (acute exposure), 96 h (chronic exposure)	pH 5.8, 96 h: Significant ↗ in AChE activity in brain	[50]
AlCl_3_	*Danio rerio*	*In vitro*, brain homogenate of adults (6–8 months) *Conc.:* 50, 100, 250 µM *Duration:* 10 min	50 μM: Significant ↗ in AChE activity	[50]
AlCl_3_	*Danio rerio*	*In vivo*, adults (6–8 months) *Conc.:* 50 µg·L^1^ *pH:* 5.8 *Duration:* 96 h	↘ in locomotor activity, in the travelled distance, and of the maximum speed ↗ of the absolute turn angle values	[50]
AlCl_3_	*Danio rerio*	*In vivo*, adults *Conc.:* 5.69, 17.08 ppm of Al (sublethal conc.) *Duration:* 7, 14, 21, 28 d	No detected accumulation in the brain (in contrary to liver, gill, and muscle)	[54]
AlCl_3_	*Carassius carassius* *Lepomis macrochirus* *Neogobius fluviatilis* *Rutilus rutilus*	*In vivo*, adults (3–5 years old) *Conc.:* 10 mg·L^−1^ *Duration:* 45 d All 4 species were used for the experiments, except for the assessment of S100ß protein content and S100ß polypeptide fragments content in the brain (*L. macrochirus* and *C. carassius* only)	Significant ↗ in lipid peroxidation end products content in the brain in all tested species Significant ↗ in SOD activity in the brain in all tested species Astrogliosis in the brain in all tested species, Significant ↗ in GFAP content and in S100ß protein content (markers of cell response in neural tissue against toxic chemicals and different damages) ↗ in GFAP lysis protein products content (40–49 kDa) and in S100ß polypeptide fragments content (24–37 kDa)	[62]
AlCl_3_	*Salmo salar*	*In vivo*, pre-smolt *Conc.:* 0.37 ± 0.04 μmol.L^−1^ Al *pH:* 5.7 *Duration:* 2 weeks	Significant ↘ in NeuroD1 mRNA levels in the forebrain ↘ in spatial learning ability and in forebrain neural plasticity Cognitive deficit	[63]
AlCl_3_	*Channa punctatus*	*In vitro*, brain homogenate of young, middle-aged, and old individuals *Conc.:* 666 µM *Duration:* 10 min	Significant ↘ CAT activity No age dependency	[61]
AlCl_3_	*Rana arvalis* *Rana temporaria* *Rana dalmatina*	*In vivo*, embryonic and young larvae *Conc.:* 100, 200, 400, 800 μg·L^−1^ of Al *pH:* 4, 5, 6 (± 0.1) Usual pH and conc. values of acidified areas in southern Sweden *Duration:* until a week after hatching	*R. arvalis*, ≥200 μg·L^−1^, pH 5: disturbed swimming behaviour *R. temporaria* and *R. dalmatina*: no change in swimming behaviour	[64]
Al_2_(SO_4_)_3_	*Oreochromis niloticus*	*In vivo*, juveniles *Conc.:* 1, 3 μg·mL^−1^ (water treatment conc.) *Duration:* 14 d	≥1 µg·mL^−1^: significant ↗ in AChE activity in a dose-dependent manner in brain	[65]
Al_2_(SO_4_)_3_	*Oreochromis niloticus*	*In vitro*, juveniles *Conc.:* 1, 3 μg·mL^−1^ (water treatment conc.) *Duration:* 1 h	≥1 µg·mL^−1^: significant ↗ in AChE activity in a dose-dependent manner in brain	[65]
Al_2_(SO_4_)_3_	*Ctenopharyngodon idella*	*In vivo*, adults Conc.: 0.1 mg·L^−1^ of Al (maximum conc. in water to protect aquatic life; not lethal for *C. idella*) *Duration:* 12, 24, 48, 72, 96 h	In brain, ≥24 h: Significant ↘ in CAT activity, in adrenaline levels, and significant ↗ in dopamine and noradrenaline levels ≥48 h: Significant ↗ in lipid peroxidation and SOD activity in a time-dependent manner ↗ in Al conc. and BCF over time while ↘ in water	[66]
Al_2_(SO_4_)_3_	*Cirrhinus mrigala*	*In vivo*, adults *Conc.:* 5.2 (chronic exposure), 17.3 ppm (acute exposure) *Duration:* 15, 30, 60, 90 d (chronic exposure), 14 d (acute exposure)	In brain, 5.2 and 17.3 ppm, ≥14 d: accumulation of Al 5.2 ppm, ≤60 d: ↗ in uptake rate 5.2 ppm, ≤90 d: ↘ in uptake and excretion rate, ↗ in the BMF up to 90 d 17.3 ppm: Low uptake rate and BMF, and high excretion rate compared to chronic exposure	[58]
Al_2_O_3_NPs	*Oreochromis mossambicus*	*In vivo*, adults (6 ± 1.5 g, 6.5 ± 1 cm) *Conc.:* 4 mg·L^−1^ (sublethal conc.) *Duration:* 24, 72, 96 h, and 15, 30, 60 d	In brain, 96 h-30 d: Significant ↗ in weight 60 d: Significant ↘ in weight, followed by a significant ↗ after recovery period (60 d) in non-contaminated water ≥24 h: Significant ↘ in SOD, CAT, GPx and AChE activity (persistent after treatment withdrawal) ≥72 h: Significant ↘ in GSR activity (persistent after treatment withdrawal) ≥15 d: Significant ↗ in hydrogen peroxide generation level (persistent after treatment withdrawal) ≥ 30 d: Significant ↗ in lipid peroxidation level	[67]
Al_2_O_3_NPs	*Oreochromis mossambicus*	*In vivo*, adults (6 ± 1.5 g, 6.5 ± 1 cm) *Conc.:* 4 mg·L^−1^ (1/10th of LC_50_-96 h) *Duration:* 96 h, 60 d	In brain, 96 h: Pathological lesions, mild degenerative changes in all regions with mild vacuolization in neural cells (persistent after treatment withdrawal) 60 d: Severe degenerative changes along with intracellular oedema (persistent after treatment withdrawal)	[68]
Al(OH)_3_	*Rutilus rutilus* *Carassius carassius*	*In vivo*, adults Topical application of AlOH_3_ gel in the midline on the surface of the posterior telencephalon in living fish *Conc.:* NA *Duration:* 6 d	In *R. rutilus*, ≤2 d: Delayed habituation of arousal responses Brief periods of insensitivity to external stimuli Electroencephalographic seizures in which the EEG amplitude was elevated from 4–20 times its normal level Unusual, gentle lateral undulations of the body Sporadic, violent and uncoordinated motor activity In *C. carassius*, Delayed habituation of quantitatively measured cardiac arousal responses to a moving shadow stimulus compared to controls	[69]
Al^3+^ (form not specified)	*Lepomis gibbosus*	*In vivo*, adults *Conc.:* 10 mg·L^−1^ *Duration:* NA	Oxidative stress and astrogliosis in the brain astrocytes	[70]
Al^3+^ (form not specified)	*Oncorhynchus mykiss*	*In vivo*, adults (≈800 g) from an aquaculture farm in South West Scotland *Conc.:* 8–9 mol·L^−1^ (mean conc. occurring in farm water) *Duration:* 2 years	Accumulation in the cerebrovascular endothelium of the BBB and in the telencephalon	[55]

↘ for decrease, ↗ for increase, d for day, hpf for hours after fertilization.

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
