# Peer review of "Effects of Aluminium Contamination on the Nervous System of Freshwater Aquatic Vertebrates: A Review"

_ijms, 2021, doi:10.3390/ijms23010031_

Round 1

Reviewer 1 Report

In this manuscript authors reviewed the recent studies on the assessment of aluminium toxicity on the nervous system of freshwater aquatic vertebrates. Overall, this is a clear, succinct ad well-written review article. I recommend this article for publication in Int. J. Mol. Sci after minor revision.

There are few points that need to be addressed in the paper:

  1. Authors need to provide a graphical abstract to present an overview of the review.
  2. The abstract does not provide a concise and complete summary, the Abstract must be rewrriten.
  3. The information in the table should better presented. For example, I recommend to add new columns for exposure conditions, with Lifespan Stages (embryos, larvae and adults), concentrations (ug L-), Exposure Period with Acute or chronic exposure, as well as the effects should be presented more concise.
  4. The sentence ,,In conclusion, further studies are needed to better characterize Al neurotoxic effects during whole developmental processes with the determination of the critical periods of time, duration, and the quantities that threaten freshwater life." need to be detailed.

Author Response

First of all, we would like to thank the reviewer #1 for the work he made. All the remarks were considered to improve the quality of our review. Modifications were made through the manuscript and appear in red text in the document.

  1. A graphical abstract was added.
  2. The abstract was rewritten.
  3. All the data concerning the lifespan stages, the exposure periods and the effects were in our tables. Since these data were not available in another review, it appears important to let them in the table. Moreover, as you can see in the manuscript, a significant work of formatting was made in order to present all these data. Add new columns will not allow to keep this format. 
  4. The last paragraph of the manuscript was amended.

Reviewer 2 Report

The manuscript "Effects of aluminium contamination on the nervous system of freshwater aquatic vertebrates: a review" provides an up-to-date and important information on the effects of aluminium on aquatic vertebrates. This review compiles scientific data on the neurotoxicity of aluminium and identifies biomarkers of aluminium exposure for aquatic environment biomonitoring in freshwater aquatic vertebrates. It also poses the question of accurate aquatic vertebrates as model organisms that could complement the biological data relating to the developmental aspect.  This data is relevant for a broad public since this the freshwater pollution with heavy metals increased in the last decades. It is perfectly clear to anyone familiar with this field of work that it had to be a great effort for authors to plan and conduct such a study, and I have a great appreciation for this. The manuscript is written in clear language and the background provides sufficient literature review. Overall, a good read.

I fully support the publication of this paper in International Journal of Molecular Sciences.

Author Response

First of all, we would like to thank the reviewer #2 for the work he made. Some points were corrected, changed and / or amended throughout the manuscript. This appears in red text in the document.